# Citizen Stance towards Mandatory COVID-19 Vaccination and Vaccine Booster Doses: A Study in Colombia, El Salvador and Spain

**DOI:** 10.3390/vaccines10050781

**Published:** 2022-05-15

**Authors:** Isabel Iguacel, Juan Pablo Álvarez-Najar, Patricia del Carmen Vásquez, Judith Alarcón, María Ángeles Orte, Eva Samatán, Begoña Martínez-Jarreta

**Affiliations:** 1Faculty of Health Sciences, University of Zaragoza, 50009 Zaragoza, Spain; jpalvarez@unillanos.edu.co; 2Instituto Agroalimentario de Aragón, 50013 Zaragoza, Spain; 3Instituto de Investigación Sanitaria Aragón, 50009 Zaragoza, Spain; mjarreta@unizar.es; 4Centro de Investigación Biomédica en Red de Fisiopatología de la Obesidad y Nutrición, 28029 Madrid, Spain; 5Faculty of Medicine, University of El Salvador, 2511-200 San Salvador, El Salvador; patricia.vasquez3@ues.edu.sv; 6Faculty of Medicine, University of Zaragoza, 50009 Zaragoza, Spain; 569059@unizar.es (J.A.); 689671@unizar.es (M.Á.O.); 190958@unizar.es (E.S.)

**Keywords:** SARS-CoV-2, vaccination, booster, mandatory vaccination, attitudes

## Abstract

The infections and deaths resulting from Coronavirus Disease 2019 (COVID-19) triggered the need for some governments to make COVID-19 vaccines mandatory. The present study aims to analyze the position of 3026 adults in Colombia, El Salvador, and Spain regarding the possibility of making COVID-19 vaccine mandatory and the intention to be vaccinated with the booster or possible successive doses. Data from an online survey conducted from August to December 2021 among a non-representative sample of Spanish-speaking countries were collected. Multinomial Logistic Regression Models were used. A total of 77.4% of Colombians were in favor of mandatory vaccination compared to 71.5% of Salvadorians and 65.4% of Spaniards (*p* < 0.000). Women and people over 65 years of age were the groups most in favor of making the vaccine mandatory (*p* < 0.000). A total of 79.4% said they had received a third dose or would intend to receive the third dose or future doses, if necessary, compared with 9.4% who expressed doubts and 9.9% who refused to be vaccinated or did not intend to be vaccinated. Among the measures that could be taken to motivate vaccination, 63.0% and 60.6% were in favor of requiring a negative test to enter any place of leisure or work, respectively, compared to 16.2% in favor of suspension from work without pay. The acceptance of mandatory vaccination and of third or future doses varies greatly according to sociodemographic characteristics and work environment. As such, it is recommended that policy makers adapt public health strategies accordingly.

## 1. Introduction

In late March 2022, the official global death toll from Coronavirus Disease 2019 (COVID-19) had surpassed six million. On 14 April 2022, new positive cases rose up to 1,039,901 worldwide [1]. In this context, vaccination is a public health priority [2].

The COVID-19 vaccine is the most effective tool to prevent serious illness and death caused by the different variants of SARS-CoV-2 [3,4]. Despite these benefits vaccination rates have slowed down in many countries [5], probably due to a more relaxed attitude among young people toward COVID-19 and the beliefs that new variants (such as Omicron) are less severe than previous ones (such as Delta) [6].

Among the possibilities that governments have envisaged to increase COVID-19 vaccination rates is the requirement for proof of a negative COVID-19 test for certain activities or making vaccinations mandatory for either all individuals or for certain health workers and other high-risk groups [7].

There are only a few countries in the world that have decided to make the vaccine mandatory for all adults or for certain sectors of the population. However, these measures have aroused the reluctance of some groups and have not always led to an increase in the number of vaccines administered [1]. While some studies have reported that more than half of participants would support mandatory vaccinations in Greece or Germany [8,9]. other investigations have shown very low levels of support for mandatory vaccination in the UK and the USA [10,11].

Although the benefits of the COVID-19 vaccine are largely known [12] the highly contagious omicron variant is widely believed to be less severe than previous forms of the virus, creating a new wave of hesitancy to become vaccinated and against a mandatory COVID-19 vaccine [13]. Thus, the present study aims to analyze the predictors of the possibility of accepting to make the COVID-19 vaccine mandatory and the intention to be vaccinated with a booster or with possible successive doses in Colombia, El Salvador and Spain.

## 2. Materials and Methods

### 2.1. Study Design and Participants

Data were collected from an online anonymous survey conducted from August to December 2021 (Appendix A). This cross-sectional study included a convenience sample of the adult population in three countries: Colombia, El Salvador and Spain.

We used Google Forms, an online survey platform (see Appendix A), to publish the questionnaire, and the link generated was then shared via social networks such as Facebook, Twitter and WhatsApp. Furthermore, healthcare professionals who work at university hospitals of three cities in Spain (Zaragoza, Logroño, and Murcia), in El Salvador (San Salvador) and in Villavicencio (Colombia) were contacted via email with the support of the Health Research Institute of each city to get the maximum sample of this professional group. The interviewees visited the URL on their electronic devices to answer the questionnaire. The inclusion criteria were individuals who (1) were 18 years old or older, (2) voluntarily agreed to participate in the online survey and (3) were able to read and complete the self-administered questionnaire independently.

After excluding those participants who did not meet the inclusion criteria (*n* = 32), a total of 3026 participants were finally included in the present study. The sampling technique in this dataset is convenience sampling.

This survey data was approved by different Ethics Committees in all the countries included: in Spain, the Ethics Committee of Aragon (CEICA), (No. C.I. 422 PI21/195 and Ethics Committee of the University of Murcia (ID: 3449/2021); CNEIS/2021/40 in El Salvador and the ESE Centro de Salud la Candelaria La Capilla (Boyacá), NIT.820003193-1 for Colombia. The survey was conducted according to the Declaration of Helsinki.

### 2.2. Measures

#### Sociodemographic and Vaccine-Related Information

The study questionnaire was divided into three main sections. The first section was designed to collect general information about the participants such as gender, age group, migrant status, socioeconomic status (SES) information that included study level and profession. The second section focused on measuring intention to vaccinate against COVID-19 (first, second and a vaccine booster dose or possible future subsequent doses) and the citizen stance towards mandatory COVID-19 vaccination. The third section focused on the COVID-19 vaccine-related data such as type and date of COVID-19 vaccine, possible side effects experienced after the first and the second jab, timing and duration of the side effects, medication to prevent or relieve post-vaccination side effects and information received before getting the vaccine about possible side effects and co-morbidities.

In the present study, participants were required to state the following information: (1) Sociodemographic information such as gender (male vs. female); (2) age (recategorized in the following group: 18–25, 26–35, 35–45, 45–55, 56–65 and 65+ years old); (3) country where they lived (Colombia, El Salvador, Spain); and (4) educational level (participants were asked to indicate their highest level of education. The response categories for each country were coded according to the International Standard Classification of Education (ISCED 1997) and re-categorized into three categories: low (ISCED level 0–2), medium (ISCED level 3–4), and high (ISCED level 5–6) [14]; (4) healthcare setting (Yes: studies related to health sciences -i.e., nursing, medicine- or working in a sanitary sector vs. No); (5) forefront COVID-19 (Yes: individuals who stated they were at the front-line such as medical doctors-MDs- or nurses in direct contact with COVID-19 patients vs. no); (6) occupation status (auxiliary or technicians, Commercials, administrative staff in hospitals and health centers, nurses, MDs; non-working participants such as students/unemployed/retired; other health professionals different from MDs or nurses such as biologists, veterinarians, psychologists and non-sanitary professionals). For statistical reasons commercials and administrative staff were categorized in the same group; (7) vaccination status (No, because I was not prioritized yet for vaccination, No, because I had the COVID-19, No, because I refused to, Yes, but only partially with one-dose, Yes, but only partially because I had the COVID-19, Yes and I got the necessary doses). For statistical purposes, vaccination status was categorized as in the following groups: No, because I did not want to, No, due to other reasons (i.e., because I was not prioritized yet for vaccination or because I had the COVID-19), Only partially (i.e one dose out of two and I am waiting for the second dose or I had the COVID-19 and I had already one dose), Yes, and I got the necessary doses; (8) Violation of human rights in mandatory vaccines (possible responses among individuals were: yes, if the thought mandatory COVID-19 vaccines would violated human rights; do not know or no); (9) possibility of having a third/booster or subsequent doses if offered (the possible responses were Yes vs. I would have doubts vs. No); and (10) believes in mandatory COVID-19 vaccination (responses options were: No, vaccines should not be mandatory for anyone, Yes, vaccines should be mandatory but only for some sectors, for example for nursing home assistants or health workers, Yes, vaccines should be mandatory for everyone).

### 2.3. Statistical Analysis

For descriptive analyses, numbers and percentage were used to detail sample characteristics. Percentages and chi-square tests were used to evaluate the sample characteristics according to their attitude towards mandatory COVID-19 vaccination.

Subsequently, we conducted Multinomial Logistic Regression Models to study the predictor factors about the acceptance (1) of making COVID-19 vaccination mandatory (with the following options: Yes, vaccines should be mandatory for everyone, Yes, vaccines should be mandatory for some sectors, and reference: No, vaccines should not be mandatory for anyone) and (2) to be vaccinated with a booster and additional subsequent doses (the possible responses were Yes, I would have doubts and reference: No). For the above regression, adjusted odds ratio (aOR) and the respective 95% confidence intervals (CI) were estimated. All analyses were performed using SPSS version 26.0 (IBM Corporation, New York, NY, USA). The alpha level was set at 0.05, and *p* < 0.05 was considered statistically significant. All models were adjusted for gender and age except for the variables age (adjusted for gender) and gender (adjusted for age).

## 3. Results

Table 1 presents the characteristics of the sample of this study. Most of the participants included were females (66.7%), aged 18–25 years (40.2%), with a high educational status (i.e., graduated or post-graduated) (50.9%) and reported to be fully-vaccinated (78.4%). A total of 37.1% reported to be in a healthcare setting (i.e., working in a hospital, students from Medicine or Nursing). Among health professionals, 4.6% were technicians, 8.7% were nurses, 7.3% MDs and 4.0% other health care professionals different from nurses and MDs such as veterinarians, nutritionists, psychologists, biologists.

Regarding the country where participants currently lived, 40.3%, of the participants reported to be from Colombian, 14.0% from El Salvador and 45.7% from Spain.

Concerning the COVID-19 vaccination status, 4.7% stated they did not get vaccinated at all because they refused; 7.1% reported not to be vaccinated due to other reasons (i.e., the individual; 9.9% did had not been prioritized yet, had the COVID-19); 9.9% had just one dose and 78.4% had both doses.

A total of 80.7% stated they would get a third dose or a possible consecutive dose of the COVID-19 vaccine; 9.4% would have doubts and 9.9% refused it.

A total of 71.1% were in favor of having a mandatory vaccine for everyone, 7.2% just for some sectors (such as health care professionals) and 21.7% were against.

A total of 63.1% of the participants stated that making the vaccination mandatory was not a violation of human rights, 28.1% thought it was, and 8.8% reported to did not know how to respond.

Mandatory COVID-19 vaccination was supported in a higher percentage by the following subgroups: particularly 77.8% of older adults (*p*-value < 0.000), 73.4% of females (*p*-value < 0.000), 73.2% of those who were in healthcare setting (*p*-value = 0.132), 72.5% with a low and 74.2% with a medium education (*p*-value < 0.030), 77.4% of Colombians (*p*-value < 0.000), 77.9% of those who reported to be in the forefront of the COVID-19 (*p*-value < 0.000), 84.9% of those who were commercials or administrative staff working in a hospital (*p*-value = 0.001), 74.4% those who got all the vaccines (*p*-value < 0.000), and 87.8% of those who though mandatory vaccinates was not a violation of human rights (*p*-value < 0.000), approved mandatory COVID-19 vaccination.

Table 2 indicates the percentage of agreement with possible measures to promote vaccination. Most people were in favor of asking PCR test or antigens to accede to leisure activities such as restaurants, cinema, theatre, gym, to travel (63.0%) or to accede to official buildings or workplaces (60.6%). However, most of the respondents disagreed in suspending from work without pay those who refused to vaccinate (67.7%).

Predictor factors about the acceptance of making COVID-19 vaccines mandatory (reference: no) are illustrated in Table 3. The results from Multinomial Logistic Regression Models adjusted for age and gender indicate that Colombians were statistically more in favor of making vaccines mandatory for everyone (aOR = 1.68, 95% CI: 1.36–2.08) and less in favor for just some sectors (aOR = 0.32, 95% CI: 0.20–0.51) compared to Spanish people. Those participants who reported to have a low (aOR = 1.40, 95% CI: 1.11–1.76) or medium education (aOR = 1.45, 95% CI: 1.15–1.82) had more chances to agree in making vaccines mandatory for everyone than participants with a high educational status. 

Those who reported not to be in the front-line against COVID-19 had lower odds of being in favor of mandatory vaccines (aOR = 0.67, 95% CI: 0.53–0.85) than those who were.

Moreover, those who were working as commercials or had an administrative job in a hospital (aOR = 2.65, 95% CI: 1.09–6.43), MDs (aOR = 1.37, 95% CI: 1.09–1.70), other health professionals different from MDs or nurses (aOR = 1.82, 95% CI: 1.08–3.08) and those who did work such as students, unemployed or retired (aOR = 1.36, 95% CI: 1.09–1.70) were more likely to be in favor of mandatory vaccines than those with a non-sanitary job. 

Participants who manifested having refused the vaccination were less likely to accept a possible mandatory vaccine (aOR = 0.03, 95% CI: 0.02–0.05) compared to those who got the necessary doses. Finally, those who thought mandatory vaccines did not constitute a violation of human rights were more likely to accept a possible mandatory vaccine for everyone (aOR = 16.04, 95% CI: 12.64–20.35) or for some sectors (aOR = 2.88, 95% CI: 1.98–4.20) compared to those who thought mandatory vaccines constitute a violation of human rights. Similarly, those who were not sure about mandatory vaccines being a violation of human rights were more likely to accept a possible mandatory vaccine for everyone (aOR = 2.96, 95% CI: 2.11–4.16) or for some sectors (aOR = 3.21, 95% CI: 1.98–5.17) compared to those who thought mandatory vaccines constitute a violation of human rights.

When analyzing gender (adjusted for age) and age (adjusted for gender), men were less likely to be in favor of mandatory vaccines (aOR = 0.77, 95% CI: 0.63–0.94) than women. As the age increased by one year, the probability of supporting mandatory vaccination increased by 1.01 units (aOR = 1.01, 95% CI: 1.00–1.02).

Predictor factors of agreeing to be vaccinated with a third and additional subsequent doses if offered (reference: no) are displayed in Table 4.

The results from Multinomial Logistic Regression Models adjusted for age and gender revealed that Colombians were six times more likely respectively to get vaccinated with a third and additional subsequent doses (aOR = 6.16, 95% CI: 4.21–9.03) compared to Spaniards. Similarly, Salvadorians were one and a half times more likely to receive a COVID-19 vaccine booster (aOR = 1.54, 95% CI: 1.06–2.24) compared to Spaniards.

Medium educational status was associated with a greater predisposition to be vaccinated with a third and subsequent doses (aOR = 1.64, 95% CI: 1.24–2.16) compared to those with a higher education. 

Those participants who were not in the front-line of the COVID-19 were more likely to express doubts regarding the vaccination (aOR = 1.63, 95% CI: 1.00–2.65) compared to those who manifested to be in the front-line.

Individuals who were in a Health Environment were more likely to accept being vaccinated with a third and additional subsequent doses (aOR = 1.64, 95% CI: 1.24–2.16) compared to those who were not.

Regarding occupational status, nurses (aOR = 2.44, 95% CI: 1.33–4.46) and MDs (aOR = 2.34, 95% CI: 1.22–4.45) were more than twice times more likely to get vaccinated with a third and additional subsequent doses compared to non-sanitary.

Participants who manifested having refused the vaccination (aOR = 0.01, 95% CI: 0.00–0.01) and those who vaccinated only partially (aOR = 0.46, 95% CI: 0.30–0.71) were less likely to get a third and additional subsequent doses compared to those who got the necessary doses. Finally, those who thought that mandatory vaccinations were not a violation of human rights were more likely to agree to be vaccinated with a third and subsequent doses (aOR = 8.65, 95% CI: 6.44–11.61) or express doubts about a possible violation of human rights (aOR = 3.07, 95% CI: 1.89–4.96) compared to those who thought mandatory vaccines constitute a violation of human rights. 

When analyzing gender (adjusted for age) and age (adjusted for gender), men expressed less “I would have doubts” about being vaccinated with a third and additional subsequent doses (aOR = 0.64, 95% CI: 0.43–0.94) than women. As the age increased by one year, the probability of supporting mandatory vaccination increased by 1.01 units (aOR = 1.01, 95% CI: 1.00–1.02).

No other statistically significant associations were found. 

## 4. Discussion

The present study aimed to analyze citizens’ stance towards mandatory COVID-19 vaccination and vaccine booster doses in three different countries, Colombia, El Salvador and Spain. We included data on 3026 adults from an online survey conducted from August to December 2021. We found that during that period most of the participants were in favor of a mandatory COVID-19 vaccination, particularly women, older participants, those with a low or medium educational status, Colombians, those who were at the front line in the fight against the COVID-19, those who were working as commercials or had an administrative job in a hospital, MDs, other health professionals different from MDs or nurses and those who did work such as students, unemployed or retired and those who thought mandatory vaccines did not constitute a violation of human rights.

Regarding the beliefs about the willingness to receive a COVID-19 vaccine booster or possible future subsequent doses, the following groups presented a greater predisposition to be vaccinated with a third and subsequent doses: older patients, Colombians, those participants with a medium educational status, those working in a health environment, nurses, MDs, those previously vaccinated with two doses and those who thought mandatory vaccines did not constitute a violation of human rights.

Contrary to our expectations, a high percentage of the population agreed to include COVID-19 mandatory vaccines for all the population. Most of the studies analyzing the attitudes towards mandatory COVID-19 vaccination were conducted in European Union member states. These studies showed lower percentages of respondents who were in favor of mandatory COVID-19 vaccination compared to our study, particularly, around 50% in Germany [15], 43% in France [16], 27.8% in Cyprus [17]. We are not aware of studies carried out in other Latin-American countries, hence the differences found regarding the percentage.

It should be taken into account that during the data collection the incidence and/or deaths due to COVID-19 was very high in Colombia, El Salvador and Spain, so the population was probably more prone to vaccination and mandatory vaccines [18]. According to the latest data as of 12 April 2022, Spain is one of the countries with the highest percentage of population vaccinated, while Colombia and, particularly, El Salvador have lower percentages. Particularly, at the end of the data collection, in December 2021, in Spain, 80% of the population had two doses, 65.1% in El Salvador, and 55.7% in Colombia. Regarding the third dose or booster, 29% of Spaniards had a booster, 15.1% in El Salvador, and 6.5% in Colombia [19]. However, a decrease in the willingness to be vaccinated with the third dose has been seen in the population of most countries worldwide. In fact, at present, barely half of the population in Spain have vaccine booster shots as a result of lower incidence, as official data have showed [3]. Additionally, while COVID-19 vaccines have successfully reduced the rates of infections, severity, hospitalization, and mortality among different populations [20], the vaccine effect on reducing transmission appears to be minimal in the context of omicron and delta variant circulation [21]. During data collection, the most prevalent variants were Delta in El Salvador and Spain and Mu in Colombia [22]. The emergence of new variants of concern have led to a reduced effectiveness of available vaccines against COVID-19 [23]. In fact, this may be also another reason for the slowdown registered in most countries regarding to the third dose. Although new studies have suggested that COVID-19 vaccines are slightly less effective against new variants, COVID-19 vaccines still appear to provide protection against severe COVID-19 [24,25,26]. 

There were no gender differences in the willingness to get vaccinated in the present study, although women agreed to support a policy of mandatory vaccination. These results are in line with the fact that women have been found to be more likely to perceive SARS-CoV-2 as a very serious virus and to agree and comply with restraining measures [27]. However, the study of Graeber et al., conducted in Germany, found that women were less willing to get vaccinated and to support a policy of mandatory vaccination [15].

Regarding age, our study found a greater probability of supporting mandatory vaccination when age increases, in line with other studies [15,16,17].

Even though acceptance of COVID-19 vaccination has been found to be positively related with a higher educational level [28], in our study those with a medium education were the participants more willing to receive a booster or possible future doses. On the other hand, participants with a low or medium education status were the ones who were more probable to support COVID-19 mandatory vaccination compared to those with a high education status. This can be explained by the fact that people with a high education might be more critical with extreme actions such as mandatory vaccines. 

Finally, in our sample, an unexpected high number of participants expressed support for mandatory vaccination. Nonetheless, when participants were asked about measures to enforce it, namely suspension from work without pay, the majority rejected these measures and accepted less stringent ones, such as requiring PCR and antigen testing for any leisure or work activity.

To promote COVID-19 vaccination, it would be advisable for the authorities to point out the safety of current vaccines. Side effects of COVID-19 vaccines such as fever, headache, fatigue, and pain at the injection site reported have been commonly reported worldwide. However, most side effects have been mild or moderate and improved within a few days of vaccination [29,30]. In addition, anaphylaxis rates associated with COVID-19 vaccines were comparable to those of other vaccines [31]. Even though previous studies have claimed the safety of being vaccinated for COVID-19 during pregnancy [32,33], some of the respondents avoided COVID-19 vaccination due to a pregnancy status [34]. To avoid future rejections in COVID-19, authorities should give strong messages about the safety and efficacy of vaccines. On the other hand, since vaccines do not prevent infection and vaccinated people can still contract the disease, it is also necessary to inform about these issues so as not to provoke a refusal effect.

To our knowledge, no studies have been conducted in Colombia, El Salvador or Spain concerning the support of mandatory vaccines for COVID-19 and intentions to get vaccinated with a booster of future doses if needed. Colombians and Salvadorians were more in favor of having a mandatory vaccine for COVID-19 compared to Spaniards. Although many countries worldwide have exposed the idea of a mandatory vaccination, at some point only a few countries have decided to apply this measure. There are some countries such as Italy and Greece where mandatory vaccines have been established for those considered at high risk (i.e., over the age of 60 in Greece and those over the age of 50 in Italy). Other countries, such as Tajikistan, Turkmenistan, Indonesia, Micronesia and Ecuador established a mandatory vaccine for all adults. Although Austria approved a mandatory vaccine for all adults, the government suspended this measure. In practice, despite mandatory vaccination, the vaccinated population in these countries is not significantly higher than in those countries where vaccination is not mandatory [3].

In line with our hypothesis, those who thought mandatory vaccines did not constitute a violation of human rights were more likely to accept mandatory vaccines and to have a greater predisposition to be vaccinated with a third or subsequent doses.

Finally, some studies have reported differences in the willingness to accept vaccination according to health occupation. Although other studies have shown that physicians would have a much higher uptake of vaccination than nurses and other professionals [35], in our study both healthcare professionals reported a significantly higher predisposition towards a third or possible successive doses compared to non-health professionals. However, no such higher predisposition was found in health care assistants, technicians, or other health care professionals.

## 5. Strengths and Limitations

As previously mentioned, we are not aware of any study that had included participants from three different countries (Colombia, El Salvador or Spain) with two objectives: to analyze (1) the support for mandatory vaccines for COVID-19 and (2) intention to get vaccinated with a booster or future doses if needed. Additionally, we included a sample size of 3026 individuals. 

Nonetheless, this research is not without limitations. Firstly, this study is not random and therefore is not representative of either Colombia, El Salvador or Spain. Secondly, there are some groups that could be underestimated, in part due to the collection method used (i.e., males represented 33.3% of the sample, participants with a low education were just 23.2% vs. those with a high education, 50.9% and older adults (<65 years old represented just 3.9% of the sample). Indeed, according to official data in 2020, 40% of the population between 25 and 65 years old had a tertiary education (higher education) in Spain, 25% in Colombia and around 6% in El Salvador [36]. Hence, the extrapolation of these results can be difficult. In fact, although online questionnaires are simple tools that can offer advantages such as the access to different types of population and prompt answers, some questions that can arise when auto-filling the questionnaire and could be responded in a face-to-face interview are difficult to address in online surveys. Also, it should be borne in mind that at the time of data collection, the booster dose had not been considered for most of the population and acceptance of the booster was hypothetically discussed. Moreover, there were some variables that could in fact measure the same dimension as the variable being explained (such as to be in favor of mandatory vaccines and the position about a possible violation of human rights when using mandatory vaccines). Finally, in this study, a direct question about COVID-19 convalescence and the strength of symptoms was not addressed. This might be also an important factor influencing the attitude towards vaccination. Consequently, results should be interpreted and considered on the bases of all the above. 

## 6. Conclusions

The present study aimed to investigate citizens’ attitudes towards a hypothetical mandatory COVID-19 vaccination and the intention to be vaccinated with a future booster or successive doses in three different countries, Colombia, El Salvador and Spain. The results collected between August and December 2021 revealed that most participants were in favor of mandatory vaccination against COVID-19, especially women, older participants, those with a low or medium level of education, Colombians and those on the frontline of the fight against COVID-19. However, only 16% were in favor of suspension of employment and pay. Approximately 80% expressed a willingness to receive a third or possible successive dose, compared to almost 10% who rejected it and another 10% who were hesitant. A greater willingness to be vaccinated with a third and subsequent doses was found among older patients, Colombians, participants with an average level of education, those working in a health care setting, MDs and nurses, those who had previously been vaccinated with two doses, and those who thought that compulsory vaccination was not a violation of human rights. Acceptance of mandatory vaccination and of third or subsequent doses varies greatly according to socio-demographic characteristics and work environment. It is therefore recommended that policy makers adapt public health strategies accordingly.

## Figures and Tables

**Table 1 vaccines-10-00781-t001:** Sample characteristics according to their attitude towards mandatory COVID-19 vaccination: Responses options were: No, vaccines should not be mandatory for anyone (*n* = 657, 21.7%), Yes, vaccines should be mandatory but only for some sectors, for example for nursing home assistants or health workers (*n* = 217, 7.2%), Yes, vaccines should be mandatory for everyone (*n* = 2152, 71.1%). (*N* = 3026).

*N* = 3026	N (%)	No (%)	Yes, for Everyone (%)	Yes, but Only for Some Sectors (%)	*p*-Value
**Age (in years) ***					
18–25	1215 (40.2)	21.2	70.8	8	**<0.000**
26–35	539 (17.8)	26.2	68.8	5	
36–50	780 (25.8)	23.2	70.5	6.3	
51–64	375 (12.4)	16	74.7	9.3	
>65	117 (3.9)	14.5	77.8	7.7	
**Gender ***					
Male	1008 (33.3)	23.9	66.6	9.5	**<0.000**
Female	2018 (66.7)	20.6	73.4	6	
**Healthcare setting**					
No	1902 (62.9)	22.5	69.9	7.6	0.132
Yes	1124 (37.1)	20.4	73.2	6.4	
**Education ***					
Low	703 (23.2)	19.3	72.5	8.1	**0.03**
Medium	782 (25.8)	19.7	74.2	6.1	
High	1541 (50.9)	23.8	68.9	7.3	
**Country ***					
Colombia	1219 (40.3)	18.2	77.4	4.2	**<0.000**
El Salvador	425 (14.0)	21.4	71.5	7.1	
Spain	1382 (45.7)	24.9	65.4	9.7	
**Forefront COVID-19 ***					
No	2024 (66.9)	23.7	68.3	8	**<0.000**
Sometimes	381 (12.6)	18.1	74.8	7.1	
Yes	621 (20.5)	17.4	77.9	4.7	
**Occupation ***					
Students/Unemployed/Retired.	1042	20.2	70.7	9	**0.001**
Administrative staff	20 (0.7)	10	85	5	
Auxiliary or technician	138 (4.6)	18.8	76.8	4.3	
Commercial	33 (1.1)	12.1	84.8	3	
Nurse	264 (8.7)	21.2	74.6	4.2	
Medical Doctor (MD)	222 (7.3)	18.9	72.1	9	
Other health professionals different from MDs or nurses	121 (4.0)	15.7	81.8	2.5	
Non-sanitary	1186 (39.2)	25	68.1	6.8	
**Vaccination status ***					
No, because I was not prioritized yet	136 (4.5)	19.1	75.7	5.1	**<0.000**
No, because I had the COVID-19	78 (2.6)	20.5	74.4	5.1	
No, because I refused to	142 (4.7)	84.5	10.6	4.9	
Only partially (one-dose)	203 (6.7)	18.9	72.1	9	
Only partially because I had the COVID-19	96 (3.2)	25	63.5	25	
Yes, and I got the necessary doses	2371 (78.4)	18.1	74.4	7.4	
**Violation of human rights in mandatory vaccines ***					
No	1909 (63.1)	7.6	87.8	4.5	**<0.000**
I do not know	266 (8.8)	25.9	58.3	15.8	
Yes	851 (28.1)	51.9	37.6	10.5	
**Third or subsequent doses ***					
No	300 (9.9)	66	27	7	**<0.000**
I would have doubts	283 (9.4)	39.9	46.6	13.4	
Yes	2443 (80.7)	14.2	79.4	6.5	

* Statistically significant difference given by the chi-square measured by the *p*-value < 0.05.

**Table 2 vaccines-10-00781-t002:** Percentage of agreement with possible measures to promote vaccination.

*N* = 3026	*N*	%
**PCR test or antigens to accede to leisure activities (i.e., restaurants, cinema, theatre, gym, to travel)**		
No	870	28.8
I don’t know	251	8.3
Yes	1905	63.0
**PCR test or antigens to accede to official buildings or workplaces**		
No	903	29.8
I don’t know	289	9.6
Yes	1834	60.6
**Suspension from work without pay**		
No	2029	67.1
I don’t know	508	16.8
Yes	489	16.2

**Table 3 vaccines-10-00781-t003:** Predictor factors about the acceptance of making COVID-19 vaccination mandatory (reference: no). Results from Multinomial Logistic Regression Models.

	Yes, for Everyone vs. No		Only for Some Sectors vs. No	
**Gender**	**aOR**	**95% CI**	**aOR**	**95% CI**
Male	**0.77**	**0.63–0.94**	1.36	0.97–1.89
Female	1.00	Reference	1.00	Reference
**Age**	**1.01**	**1.00–1.01**	1.00	0.99–1.02
**Country**				
Colombia	**1.68**	**1.36–2.08**	**0.32**	**0.20–0.51**
El Salvador	1.33	0.99–1.78	**0.62**	**0.41–0.92**
Spain	1.00	Reference	1.00	Reference
**Education**				
Low	**1.40**	**1.11–1.76**	1.46	0.99–2.16
Medium	**1.45**	**1.15–1.82**	1.09	0.72–1.65
High	1.00	Reference	1.00	Reference
**Front-line COVID-19**				
No	**0.67**	**0.53–0.85**	1.25	0.79–1.99
Sometimes	0.95	0.67–1.35	1.49	0.78–2.84
Yes	1.00	Reference	1.00	Reference
**Health Environment**				
Yes	1.13	0.93–1.37	0.96	0.68–1.35
No	1.00	Reference	1.00	Reference
**Occupation**				
Auxiliary or technician	1.43	0.9–2.27	0.93	0.36–2.40
Commercial or administrative	**2.65**	**1.09–6.43**	1.15	0.22–6.09
Nurse	1.26	0.90–1.77	0.80	0.39–1.64
Medical Doctor (MD)	**1.37**	**1.09–1.70**	1.72	0.94–3.16
Other health professionals	**1.82**	**1.08–3.08**	0.62	0.17–2.21
Students/Unemployed/Retired	**1.36**	**1.09–1.70**	**1.87**	**1.29–2.72**
Non-sanitary	1.00	Reference	1.00	Reference
**Vaccination status**				
No, because I refused	**0.03**	**0.02–0.05**	1.50	0.32–1.73
No, due to other reason	0.96	0.66–1.39	0.66	0.32–1.34
Only partially	0.81	0.59–1.10	0.85	0.51–1.44
Yes, and I got the necessary doses	1.00	Reference	1.00	Reference
**Violation of human rights**				
No	**16.04**	**12.64–20.35**	**2.88**	**1.98–4.20**
I don’t know	**2.96**	**2.11–4.16**	**3.21**	**1.98–5.17**
Yes	1.00	Reference	1.00	Reference

Statistically significant results shown in bold font; models were adjusted for age and gender except for the variables age (adjusted for gender) and gender (adjusted for age). aOR: adjusted odds ratio.

**Table 4 vaccines-10-00781-t004:** Predictor factors of agreeing to be vaccinated with a booster and additional subsequent doses if offered (reference: no). Results from Multinomial Logistic Regression Models.

	Yes vs. No		I Would Have Doubts vs. No	
**Gender**	**aOR**	**95% CI**	**aOR**	**95% CI**
Male	**0.82**	**0.63–1.09**	0.64	0.43–0.95
Female	1.00	Reference	1.00	Reference
**Age**	**1.01**	**1.00–1.02**	0.99	0.98–1.01
**Country**				
Colombia	**6.16**	**4.21–9.03**	**2.65**	**1.65–4.26**
El Salvador	1.54	1.06–2.24	1.43	0.88–2.34
Spain	1.00	Reference	1.00	Reference
**Education**				
Low	0.79	0.58–1.07	1.46	0.99–2.16
Medium	**1.69**	**1.18–2.42**	**1.85**	**1.15–2.97**
High	1.00	Reference	1.00	Reference
**Front-line COVID-19**				
No	0.81	0.58–1.12	1.25	0.79–1.99
Sometimes	0.85	0.54–1.33	1.49	0.78–2.84
Yes	1.00	Reference	1.00	Reference
**Health Environment**				
Yes	1.64	1.24–2.16	1.02	0.70–1.49
No	1.00	Reference	1.00	Reference
**Occupation**				
Auxiliary or technician	1.12	0.60–2.08	0.77	0.33–1.83
Commercial or administrative	2.26	0.65–7.88	0.34	0.03–3.78
Nurse	**2.44**	**1.33–4.46**	0.77	0.32–1.83
Medical Doctor (MD)	**2.34**	**1.22–4.45**	1.00	0.42–2.43
Other health professionals	2.07	0.90–4.76	1.32	0.46–3.80
Students/Unemployed/Retired	1.08	0.80–1.45	0.82	0.55–1.23
Non-sanitary	1.00	Reference	1.00	Reference
**Vaccination status**				
No, because I refused	**0.01**	**0.00–0.01**	**0.06**	**0.03–0.12**
No, due to other reasons	0.68	0.66–1.39	0.66	0.32–1.34
Only partially	0.81	0.59–1.10	0.85	0.51–1.44
Yes, and I got the necessary doses	1.00	Reference	1.00	Reference
**Violation of human rights**				
No	**8.65**	**6.44–11.61**	1.87	0.24–14.80
I don’t know	**3.07**	**1.89–4.96**	**4.13**	**2.35–7.24**
Yes	1.00	Reference	1.00	Reference

Statistically significant results shown in bold font; models were adjusted for age and gender except for the variables age (adjusted for gender) and gender (adjusted for age). aOR: adjusted odds ratio.

## Data Availability

The raw data supporting the conclusions of this article will be made available by the authors upon request, without undue reservation.

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
