# Peer review of "Citizen Stance towards Mandatory COVID-19 Vaccination and Vaccine Booster Doses: A Study in Colombia, El Salvador and Spain"

_vaccines, 2022, doi:10.3390/vaccines10050781_

Round 1
Reviewer 1 Report
General comment : This article addresses an important issue in three Spanish-speaking countries. However, important selection biases need to be better discussed and the analysis deepened.
Specific comment:
Abstract :
Line 23 : specify “respondents”, to not give the impression of having a representative sample
Manuscript:
Line 69: more detail is needed on how dissemination was performed via social networks
Line 149: the parenthesis is confusing
Line 159: be more precise about the analysis used to arrive at this result
Table1: please provide p-value
Table 3: precised aOR for adjusted Odds Ratio
“Male 0.77 0.63-9.94 1.36 0.97-1.89” why in bold
Occupation: do not use "other" as a reference category
The multivariate models includes all variables without regard to interactions or the fact that some variables measure the same dimension as the variable being explained. These choices should be reviewed and better explained
In addition, the analysis part says that a GEM model will be used and in the result part we see a multimodal logistic regression model.
Discussion: the discussion deserves to be deepened
Conclusion: the conclusion should be rewritten to emphasise original and important results and perspectives
Author Response
We would like to thank the reviewers for their thoughtful and constructive comments. Changes in the manuscript have been highlighted using track changes. An itemized point-by-point response to your comments is presented below.

Reviewer 2 Report
The matter presented in the manuscript is interesting and important for dealing with pandemic.
However, as the authors mentioned, the responders were targeted with social media, which for obvious reasons, results in a biased population - most probably younger, more educated, fluent in electronic media, etc. It would be interesting to compare the demographic and socioeconomic characteristics of the survey provided responses with general population data (government provided statistics) to be able to estimate how the study population is related to the general populations of the countries.
I understand that the survey did not include a direct question about COVID-19 convalescence and the strength of symptoms - I would consider it an important factor influencing the attitude towards vaccination.
There is a number of publications dealing with an attitude towards anti-SARS-CoV-2 vaccination in various populations - discuss in more detail.
Some more work is needed with the tables - please try to make the data presentation more transparent (use whole page width, maybe include OR in the same table as group sizes).
Please also be careful with the data, as some bolded OR are definitely not statistically significant (OR 0.77 CI 0.63-9.94; OR 1.36, CI 0.97-1.89).
Please also avoid using “...” in the text, it looks as if something was missing.
Moreover, it would be beneficial to include the whole survey in the supplementary material
Author Response

(The authors gave the same response as above.)

Reviewer 3 Report
Vaccination is the most effective tool to prevent any infectious disease. I fully agree with the authors that “herd immunity through vaccination is a public health priority.” However, will herd immunity to Wuhan strain protect vaccinees against Delta or Omicron strain? Some (especially vaccine developers) say “yes”, but they have no strong evidence for this.
Research was conducted in Colombia, El Salvador, and Spain in 2021. What specific variants of SARS–CoV–2 were circulating during the observations in these countries, and what particular strain was included in the vaccine? Do the authors have such information? Obviously now there are a few times fewer people who want to be vaccinated with a vaccine from the Wuhan strain than with a vaccine against, for example, Omicron. Did the interviewees know that they were offered to be vaccinated against another, no longer circulating virus? The correct answer to this question can make you look at the results of the survey from a completely different angle.
Imagine a fairly common, unfortunately, situation where a person was vaccinated twice (or even boosted) with a vaccine against the Wuhan strain, after which he became infected first with Delta, then with Omicron. Do you think this person will advocate vaccination?
I would like to ask a few questions from the position not of an expert in virology and vaccine development, but from the point of view of the layman. It seems to me that the question of what kind of virus is included in the vaccine should be of interest to ordinary people in the first place.
Do the authors know which variant is currently being vaccinated against, or do they all continue to be made based on the Wuhan strain? If the authors are able to find these data, it would be very important to compare which virus was circulating at the time of testing and which was in the vaccine.
I believe that the question “Which particular strain of vaccine is I being offered to be vaccinated with?” should be taken into account when conducting such surveys. Surely some of the respondents asked this question (perhaps highly educated people), but it is not reflected in the questionnaire.
Line 319–329. This paragraph should be included into the separate section “Limitations.” In addition, I think that the above mentioned question “Which particular coronavirus strain is being offered to be vaccinated with?” should be discussed in this new section as it is one of the limitations.
Author Response

(The authors gave the same response as above.)

Reviewer 4 Report
This is an interesting an well-conducted study. Please find below my comments to improve it.
- Abstract, first line: Please spell out the abbreviation COVID-19 the first time you use it (same for Introduction).
- Abstract: By definition, COVID-19 is a disease, you do not need to write COVID-19 disease. Please correct accordingly.
- Abstract: Please add p-values when you compare your findings.
- Introduction, first paragraph (lines 38-39): Herd immunity is questionable for COVID-19. Please revise.
- Introduction, lines 40-42: please add 1-2 references for your statements about COVID-19 benefits.
- Methods, data: pages 3 and 4 (line 94-124): There is too much detail in this section. It would appear better to reduce significantly.
- Results (lines: 159-164): please provide p-values for your comparisons.
- Table 1: please provide p-values for all comparisons in a specific column (right side of Table 1).
- Discussion: please provide the COVID-19 vaccination rates in the general public in the three countries, at the time the survey was conducted.
- It would be interesting to compare your findings with those from other countries (beyond Greece and Germany mentioned in Introduction).
- Please mention the need to communicate the safety profile of COVID-19 vaccines to general public. You may want to use the following reference: Maltezou et al. Anaphylaxis rates associated with COVID-19 vaccines are comparable to those of other vaccines. Vaccine 2022;40:183-186.
Author Response

(The authors gave the same response as above.)

Round 2
Reviewer 2 Report
In my opinion, still more work is needed with the tables (3&4) - please try to make the data presentation more transparent (use whole page width, characteristics descriptions can resemble those from table 1). I am not sure, if I correctly understand the results presented in the tables - what is the actual reference? The test demands the assignment of one of the categories of a dependent variable as a reference - it looks as if a reference was set both for dependent and covariate variables. More precise description of models is needed.
There is still a number of “...” in the text, again it looks as if something was missing.
The authors declared the possibility of obtaining the survey, however, I would consider beneficial to include the whole survey in the supplementary material
Author Response
Dear reviewer,
Please find attached the point-by-point response to your comments.
Best wishes,
Dr. Iguacel
